# Genetic and Functional Differences between Duplicated Zebrafish Genes for Human *SCN1A*

**DOI:** 10.3390/cells11030454

**Published:** 2022-01-28

**Authors:** Wout J. Weuring, Jos W. Hoekman, Kees P. J. Braun, Bobby P. C. Koeleman

**Affiliations:** 1Department of Genetics, University Medical Center Utrecht, 3584 Utrecht, The Netherlands; w.j.weuring-2@umcutrecht.nl (W.J.W.); wj.hoekman@gmail.com (J.W.H.); 2Department of Neurology & Neurosurgery, University Medical Centre Utrecht, 3584 Utrecht, The Netherlands; k.braun@umcutrecht.nl

**Keywords:** Dravet Syndrome, genome duplication, SCN1A, Nav1.1, VGSC

## Abstract

There are currently seven different zebrafish strains that model Dravet Syndrome, a severe childhood form of epilepsy. These models are based on a set of duplicated genes, *scn1laa* and *scn1lab*, which are the homologs for human *SCN1A*. Disrupting one of the genes would mimic a heterozygous disease state in humans, as the paralog gene is still present. While this ‘disease-state model’ is widely accepted, there is also evidence that the function of these genes might not be completely the same. By analyzing the functional domains, we discovered several hotspots in the protein that are not conserved, indicating a functional difference. Based on this, we generated *scn1Laa* knockout zebrafish and compared their phenotype to *scn1lab* knockouts. The genetic and functional differences we discovered can have implications for the use of zebrafish as a model for Dravet Syndrome.

## 1. Introduction

Dravet Syndrome (DS) is a developmental epileptic encephalopathy caused by heterozygous loss-of-function mutations in the voltage-gated sodium channel (VGSC) gene *SCN1A*. DS has an age of onset within the first year of life and a severe disease prognosis. The past ten years, a total of seven different zebrafish models have been generated to model DS, using morpholino antisense oligomers, random mutagenesis using *N*-ethyl-*N*-nitrosourea (ENU), or CRISPR/Cas9, including the heterozygous and homozygous *scn1lab* knockout strains by our lab. These have fast-forwarded our understanding of epileptogenesis [1] and behavioral comorbidities [2], enabled rapid drug repurposing for Fenfluramine [3] and Clemizole [4], yielded the efficacy of novel VGSC subtype-selective compounds [5], and were used to test in-vivo functionality of CRISPR activation [6]. Zebrafish carry two genes for human *SCN1A* named *scn1laa* and *scn1lab* [7], which are duplicated paralog genes that were initially assumed to have similar, or even identical function. Current DS zebrafish models are based on this assumption (Figure 1).

Homozygous *scn1lab* knockout (KO) zebrafish are considered to be haploinsufficient for the Nav1.1 sodium ion channel, due to the expression of the paralog gene *scn1laa*. Therefore, under this assumption, homozygous KO of either *scn1lab* or *scn1laa* would mimic the haploinsufficiency of Nav1.1 observed in human DS patients. Six DS zebrafish models are based on disruption of *scn1lab*, which leads to locomotor hyperactivity, burst movements and epileptiform activity recorded from the brain. The drug response of *scn1lab* zebrafish models mimic that of the majority of DS patients, showing no effect, or increased epileptiform and burst movement activity after treatment with Carbamazepine or Phenytoin, and a reduction when Valproate, Stiripentol or Fenfluramine are applied. One DS zebrafish model, *scn1laa^sa1674^* is based on the paralog gene and was generated via ENU mutagenesis. This model shows a comparable phenotype to the models based on *scn1lab*.

Curiously, the spatial localization of *scn1laa* and *scn1lab* transcripts does not overlap during early development. After fertilization and until 48 h, *scn1lab* is expressed in ventral regions of the nervous system such as the hindbrain and spinal cord, while *scn1laa* is expressed in sensory neurons of the peripheral nervous system [7]. At 3 days post-fertilization (dpf) *scn1laa* and *scn1lab* are both expressed in the brain, with faint expression of *scn1lab* in the heart at 5 and 7 dpf, resembling *SCN1A* [4], although expression in the heart was not confirmed in a later study [11]. In adult zebrafish, *scn1laa* and *scn1lab* are both expressed in the brain, eye and spinal cord [12]. In addition to transcript location variations at different developmental stages, predicted amino-acid identity scores based on the initial alignments were relatively low between Scn1laa and Scn1lab, showing 67% identity in comparison to 88% for the comparison of Scn8aa and Scn8ab.

Paralog genes generally have an equal function, as they are originating from the same ancestor gene, but it is possible that they develop new functions, or discard functions that are not advantageous during evolution. While all voltage-gated sodium channels transport Na ions during the generation of action potentials, their location, action potential thresholds and interaction with beta subunits can differentiate between subtype-selective function [13]. To find out if this could be the case for *scn1laa,* we aligned Scn1laa, Scn1lab and SCN1A using Jalview and ClustalOmega. Based on the general structure of VGSC we analyzed in more detail domains such as the voltage sensor, pore region and regions involved with beta subunit binding to predict potential functional differences. We generated an *scn1laa* knockout strain using CRISPR/Cas9, performed electrophysiology experiments to measure brain activity patterns and compared the locomotor phenotype with *scn1lab*^−/−^. We summarized with implications for the DS zebrafish model as a high-throughput drug screening tool.

## 2. Materials and Methods

### 2.1. Alignment

Protein sequences for SCN1A (SCN1A-224, ENST00000674923.1) Scn1laa (scn1laa-203, ENSDART00000161648.3) and Scn1lab (scn1lab-202, ENSDART00000151247.3) were initially aligned using both ClustalOmega and Muscle via Jalview [14]. As alignment identity scores did not differ between these software (data not shown), ClustalOmega was ultimately used for percentages per protein domain. Protein domains were established using Uniprot (https://www.uniprot.org/) (accessed on 29 December 2021) [15] and final figures were generated with Biorender (www.biorender.com) (accessed on 29 December 2021).

### 2.2. Zebrafish Maintenance & Ethics Statement

All animal experiments were conducted under the guidelines of the animal welfare committee of the Royal Netherlands Academy of Arts and Sciences (KNAW). Adult zebrafish (Danio rerio) were maintained and embryos raised and staged as previously described [5]. Adult zebrafish were maintained in 4.5-L polyethylene tanks (Tecniplast) in an Aqua Schwarz holding system (Göttingen) supplied continuously with circulating UV treated filtered tap water, which was exchanged for 10–30% daily. Average water properties were: Nitrite 0.095 mg/L, Nitrate 16.7 mg/L, Chloride and Ammonium 0 mg/L, hardness 9.8 dH, pH 8.2, conductivity 460 mS, Oxygen 6.85 ppm and temperature 28.5 °C under cycles providing 14 h of light and 10 h of dark (14:10 LD; lights on 9 a.m.; lights off 11 p.m.).

### 2.3. Scn1laa Knockout Strain Generation

Knockout zebrafish were generated using CRISPR/Cas9 technology according to previously published methods [5]. In brief, 500 ng Cas9 mRNA and 150 ng sgRNA [GATGAGGTTCACCAGGTAGA] + Cas9 scaffold sequence were injected in one-cell stage zebrafish embryos. F0 founder strains were identified by outcrossing leading to F1 heterozygote strains. F2 generation 5 dpf embryos was generated by incrossing the F1 generation, and heterozygote or homozygous knockout embryos were individually validated by sequencing. Primer sequences for PCR amplification and sequencing are FW: TTTGATCCAATCCCTTATCC RV: CAACAGACCTCAGCTTCCTG.

### 2.4. Locomotor Assays

Locomotor experiments were performed according to previously published methods [5]. In brief, zebrafish larvae were placed in flat-bottom 48-well plates filled with E3 medium at 4 dpf to reduce stress from pipetting on the recording day. At 5 dpf, movements were tracked in an automated tracking device (ZebraBox™; Viewpoint, Lyon, France) for 90 min, stacked in 10 min bins, of which the first 30 min were removed as habituation time for the locomotor chamber. Threshold parameters for the burst movement protocol were freezing 1, sensitivity 8 and burst 50 resulting in a burst movement cut-off value of 50 mm/s. A total of 12 animals were used per group. Data did not pass the test for normality, therefore the Mann–Whitney U test was used for statistical analysis.

### 2.5. Local Field Potential Recordings

Local field potential recordings of zebrafish brain activity were performed according to previously published methods [5]. In brief, each larva was exposed to 10 μM d-Tubocurarine pentahydrate as a muscle relaxant for 2 min, in order to reduce electro-mechanical artefacts caused by physical twitching, and was then embedded in 1.5% low-melting point agarose. Next, a silver wire carrying glass electrode connected to a high-impedance amplifier, filled with 1 mM NaCl was directed on top of the forebrain of 5 dpf larvae. Recordings were performed in current clamp mode using the DAGAN EX-1 amplifier, national instruments 6210 USB digitizer and LabscribeNI at 1000 samples per second. A 50 Hz notch filter was applied and 2.5–100 Hz signals were further analyzed. The fast-Fourier transform function in Labscribe software was used to generate power spectra of 40 s traces. Each recording lasted 13 min, of which the first was used to place the electrode and removed from the final figures. Three full-length recordings with marked epileptiform events are added as Appendix A.

## 3. Results

### 3.1. Structural Alignment

Previous alignments of Scn1laa and Scn1lab with human SCN1A showed only overall similarity without much focus on domains with functional importance. We first performed multiple sequence alignment of Scn1laa or Scn1lab versus SCN1A, that showed predicted protein-wide identity scores of 67% for Scn1laa and 77% for Scn1lab, confirming previous findings [7,12]. This alignment result was used to calculate the percentages of amino acid conservation for the 58 different protein domains of SCN1A as defined in Uniprot. In more detail, out of the 58 protein domains, 12 showed more than 20% difference in conservation between Scn1laa and SCN1A when compared to the conservation between Scn1lab and SCN1A (Figure 2). These low-conserved domains are spread over the ion channel and include transmembrane (TM) segments, intracellular linkers, extracellular loops and the N-terminus. Important regions for the primary function of VGSC are the S4 voltage sensor, S5 and S6 pore-lining segments, the S5–S6 pore-forming loop X and the S2–S3 cytoplasmic linker (inactivation gate). Two of these—the S4 voltage sensor segment in domain II and cytoplasmic linker E in domain IV—exceeded more than 20% difference.

### 3.2. Scn1laa Knockout Phenotype

Heterozygous and homozygous *scn1laa* knockout zebrafish larvae (*scn1laa^+/−^* and *scn1laa^−/−^)* were generated using CRISPR/Cas9 and carried a 7 bp deletion in exon 9, leading to a premature stopcodon that predicts a complete loss of function (Figure 3A). All experiments described below were performed with animals at 5 dpf, comparable to previous publications using DS zebrafish. *Scn1laa^−/−^* larvae do not share the hyper-pigmentation phenotype, nor the absence of an inflated swim bladder with models based on *scn1lab* (Appendix A). Local Field Potential (LFP) recordings were used to evaluate the presence of abnormal brain activity as described before (4). LFP recordings of *scn1laa^−/−^* zebrafish showed spontaneous spike activity, which enriched both low- and high-frequency events in the fast-Fourier transform power spectra (Figure 3B). This epileptiform activity could clearly be separated from wildtype zebrafish recordings and confirmed those detected in the *scn1laa^sa1647^* zebrafish model published earlier [8]. We further tested the influence of light stimuli to induce epileptiform activity. For this purpose, zebrafish were exposed to intermittent light at 30 s intervals. This was repeated for 5 min, during which LFP recordings were continuously made. Under these conditions, *scn1laa^−/−^* showed repeated trains of biphasic spike activity that followed directly after each light stimulus, suggesting that *scn1laa^−/−^* zebrafish display a photosensitive epileptiform phenotype, which was detected in several *scn1lab* models earlier (Figure 3C).

To find out if there are differences in the locomotor profile of *scn1laa^−/−^* a locomotor assay was performed under dark conditions that measured hyperactivity and high velocity burst movements as described before (4). Surprisingly, *scn1laa^−/−^* larvae showed no alteration in overall movement activity when compared to wildtype zebrafish, unlike *scn1lab^−/−^* which are hyperactive. Burst movements, typical for *scn1lab^−/−^* and used for ASD screenings in DS zebrafish models, were also absent in *scn1laa^−/−^* highlighting a functional difference (Figure 3D).

## 4. Discussion

Summarizing the differences in early expression of *scn1laa* and *scn1lab* transcripts from previous studies and our structural alignment and phenotype data, we conclude that it is likely that *scn1laa* and *scn1lab* do not have identical functions.

We first showed that Scn1laa and Scn1lab differ substantially at the protein level. To predict whether these differences have functional consequences, we compared the amino acid conservation score for each of the 58 protein domains in VGSC. This analysis showed that sequence conservation was unevenly distributed across the protein. Twelve domains, including the functionally important voltage-sensing segment in DII and intracellular linkers, showed more than 20% sequence difference, indicating a functional difference.

Low conservation in the voltage sensor may indicate a different voltage threshold needed for channel activation and inactivation, which in turn suggests that *scn1laa* might be expressed in other cell types or at a different location within a neuron when compared to *scn1lab*. More difficult to explain are the low conservation scores in intracellular linkers and extracellular pore segments. While intracellular linkers could be associated with protein interaction intracellularly, these protein interactions are largely unknown for SCN1A and the zebrafish paralogs. A possible candidate for SCN1A is SCN1B, the beta-subunit of Nav1.1 and required for correct localization of the VGSC to the membrane [16]. However, the exact binding site of SCN1B to SCN1A is not known and since SCN1B also contains both a transmembrane and intracellular domain, this remains a topic of debate.

The extracellular loops between S5 and S6 are known to be candidates for N-glycosylation [17] and are reasonably well-conserved, showing both minorly decreased and increased conservation levels for Scn1laa, and are therefore not likely to contribute to functional or regulatory differences. Neighboring loops between S1, S2 and S3 in turn do show lower than 20% conservation for Scn1laa, but as there are no interacting partners or regulatory pathways known to act on these sites, these results are currently difficult to interpret.

Furthermore, we observed striking differences between the phenotypes of the *scn1laa^−/−^* and *scn1lab*^−/−^ zebrafish. The two different KO zebrafish do share the presence of spontaneous epileptiform activity and photosensitivity. However, the *scn1laa^−/−^* does not have the locomotor phenotype that is characteristic of *scn1lab*^−/−^ zebrafish. *Scn1lab^−/−^,* but also *scn1Lab*-based models with homozygous missense mutations, or a translational knockdown, show hyperactivity and high-velocity burst movements which are absent in *scn1laa^−/−^*. The absence of a locomotor hyperactivity phenotype indicates that *scn1laa* might be located in other neurons, other cell compartments such as the soma, axon or dendrite, or has a different action potential threshold. While all VGSC transport sodium, their spatiotemporal location is crucial for each specific subtype and determines their unique function.

It is not clear why another previously published *scn1laa* model (*scn1laa^1674^)* did show locomotor hyperactivity and this should be further investigated. It is possible that phenotypic differences arise from differences in accuracy between technologies used to introduce the mutation to generate the strain. ENU-generated models likely carry additional mutations beside the mutation that is described as causative, on top of the mutations that are crossed out after ENU treatment [18], whereas the CRISPR/Cas9 stringent sgRNA design may be more precise. Differences in chamber habituation time, possible stress from pipetting, and recording parameters can all potentially contribute to differences in locomotor phenotypes. We therefore placed larvae in a single well one day prior to the experiment, subtracted the first 30 min of data for chamber habituation, and recorded for a total of one hour.

Overall, given the increasing number of zebrafish models to study monogenic diseases, we believe that standardization of experiments is vital to rule out the majority of false negative and false positive findings. As we used standardized methods for both *scn1laa^−/−^* and *scn1lab^−/−^* strains in this study, an effective comparison could be made. With the introduction of the zebrafish epilepsy project published last year [11], a plethora of different zebrafish models can be within reach for researchers worldwide, tackling the methodology aspect as they are all generated with CRISPR/Cas9. For current drug screenings we propose that models based on *scn1lab* should be used instead and further research, for example using double heterozygote knockouts, can help better understand the functional differences. Overall, for future RNA- and DNA-based therapies that are currently in development for DS, we believe that zebrafish are no longer suitable as a model system due to the low conservation score of genes when compared to mammalian model systems.

## Figures and Tables

**Figure 1 cells-11-00454-f001:**
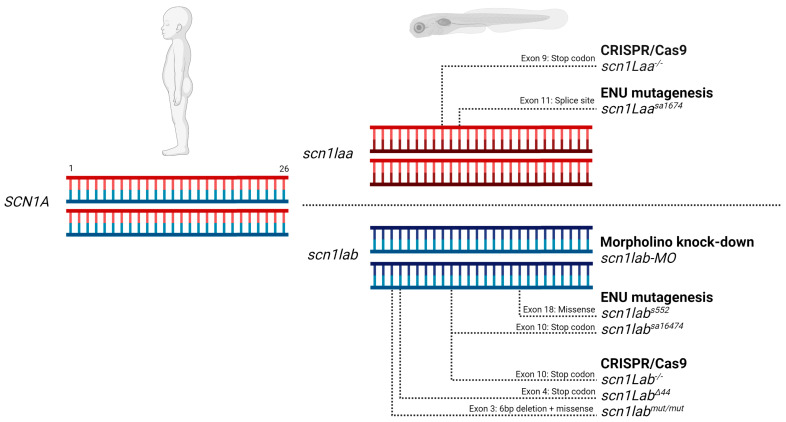
Current zebrafish models for Dravet Syndrome. SCN1A haploinsufficiency in DS patients is modeled by homozygous disruption of one of the two zebrafish paralogs. Disruption is achieved via gene knockdown, missense mutations, or gene knockout introduced by different technologies. References for each model from top to bottom; *scn1laa^−/−^* [This publication], *scn1laa^sa1674^* [8], *scn1lab* MO [3], *scn1lab^s552^* [4], *scn1lab^sa16474^* [9], *scn1lab^−/−^* [5], *scn1lab^∆44^* [10], *scn1lab^mut/mut^* [1]. Figure made with Biorender.

**Figure 2 cells-11-00454-f002:**
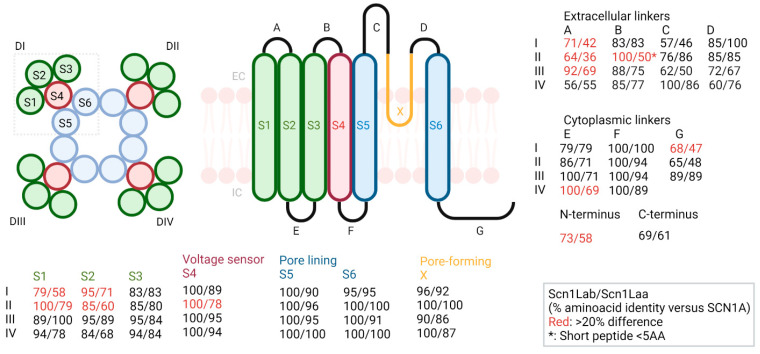
Top-down and horizontal structure of VGSC and multiple sequence alignment of Scn1laa:SCN1A and Scn1lab:SCN1A. Scores are percentage amino acid identity of Scn1lab/Scn1laa versus SCN1A. Roman letters indicate Domain I, II, III and IV, S1–S6 = transmembrane segment 1–6, A–D extracellular loops, E–G cytoplasmic linkers. Regions with more than 20% difference are highlighted in red. Short peptides are marked with an asterisk.

**Figure 3 cells-11-00454-f003:**
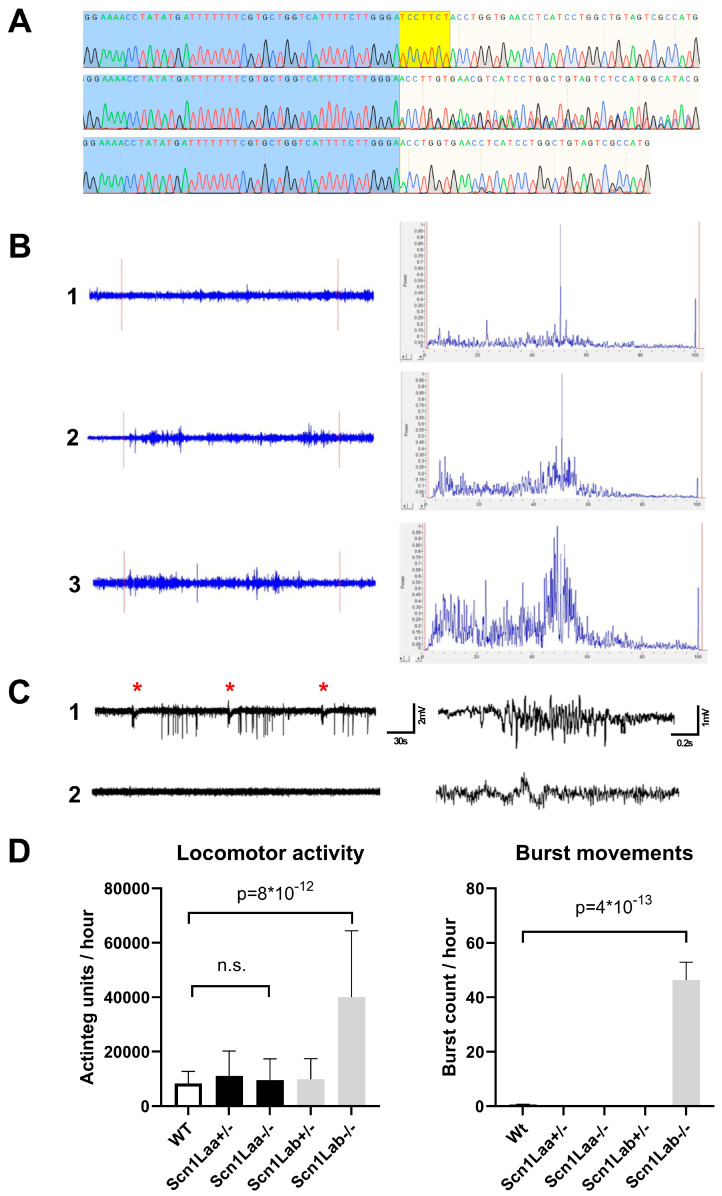
Genotype and phenotype of *scn1laa^−/−^* (**A**) Sanger trace showing wt (top), heterozygous 7 bp deletion (middle) and homozygous 7 bp deletion (bottom) strains generated for this study. The deletion is highlighted in yellow. (**B**) Local field potential (LFP) recordings of the zebrafish brain showing baseline activity of wildtype animals (1) and spontaneous epileptiform activity in *scn1Laa^−/−^* animals (2,3). Per recording, the region marked between red lines was used to generate power spectra after fast-Fourier analysis highlighting an enrichment of both low- and high-frequency signals during epileptiform events. (**C**) LFP recordings after light stimuli presented at each asterisk for *scn1laa^−/−^* (1) and wildtype (2) animals. Detailed traces are shown on the right side. (**D**) Locomotor profiles of *scn1laa* KO strains compared to wildtype and *scn1lab* KO strains [5]. *n* = 12 animals were used per group. A *p* value higher than 0.01 was considered not significant (n.s.).

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
