# Peer review of "Genetic and Functional Differences between Duplicated Zebrafish Genes for Human SCN1A"

_cells, 2022, doi:10.3390/cells11030454_

Round 1
Reviewer 1 Report
Comments to Authors
The manuscript entitled, “Genetic and functional differences between duplicated 2 zebrafish genes for human SCN1A” by Weuring et al., shows that scn1Laa-/- zebrafish has a higher difference in amino acid sequence identity to SCN1A than that of scn1Lab-/- zebrafish. Secondly, the abnormal brain activity observed in scn1Lab-/- zebrafish was observed in the scn1Laa-/- zebrafish. It would be of interest if the authors would determine whether there were differences in the spike patterns between the two strains. Lastly, the spontaneous motor activity and burst movements of the scn1Laa KO fish was like WT fish, unlike the scn1Lab KO fish. The communication is of interest but could be improved, as indicated below.
The major concern is the data in figure 3. Panel b should be c to match the flow in the text of the results. Both panels a and b are fine, however, panel c needs additional data. The authors should show the lack of spike activity in the wt fish. The manuscript would also be improved if there was a direct comparison of the KO strains. Perhaps have bar graphs comparing the number of events per duration, time between events and cumulative duration of events… This is critical since the motor activity studies indicated the scn1Laa-/- zebrafish were like wt fish. Further the text of the results describing panel c is quite limited and need to be expanded.
Figure 2, clarify the 20% difference in the box. Line 147-156 should be moved to the discussion and references added. Further the authors should discuss the possible link of N-glycosylation processing the Na+ channel to channel activity since the extracellular loops were quite different. It would also be of interest for the authors to correlate possible changes in the voltage sensor and the inactivation gate with changes in the possible brain activity differences. These brain activity differences could be better defined by conducting measurements mentioned above.
It would also be worth mentioning if future studies are planned to correlate changes in sequence of scn1Laa-/- and scn1Lab-/- with changes in channel activation and inactivation.
Minor points
Check the use of paralogues and orthologues.
Add and check references throughout such as line 89, lines 147-156, after line 173… Add references of the various models mentioned or a review. Clarify the reference for the scn1Lab-/-.
Check spelling, sentence structure, spacing, font and punctuation throughout, such as line 75, 88, 115, 117, 55-58, 123-124, 199
Author Response
Dear Reviewer 1,
We kindly thank you for reviewing our manuscript. You have pointed out several excellent improvements to our experiments of which we were able to perform several. In Figure 3 the panels are edited as requested now reading A-D. Panel B now includes a wildtype recording and two mutant recordings, together with fast-fourier transform power spectra to highlight differences between baseline and epileptiform activity which we hope you appreciate. Panel C (photosensitivity) now also has a wildtype trace added, with detailed activity displayed on the right side. Panel A and D were unaltered.
In Supplementary S2 we added full-length recordings of two 1aa KO animals which show 7-10 events per ten minutes recording time. The wildtype trace does not show any of these, further confirming that 1aa KO have a seizure-like phenotype. We were unfortunately not able to perform a more detailed electrophysiology comparison between 1aa and 1ab knockout strains as the scn1Laa KO strain died during last year CoViD lockdown, we were not allowed to continue crossing the strains until they were not fertile anymore. By comparing the 1aa KO data (7-10 events/9 minutes) with our previously published 1ab KO data [5] which shows an average of 1/minute the amount of events appears to be comparable to confirm a seizure phenotype, but given the low number of datapoints we believe this comparison is not suitable for statistical analysis.
We further added microscope pictures of 1aa and 1ab KO strains (supp.S1), clearly showing that the 1aa KO does not share the hyperpigmentation and swim-bladder defects which are present in 1ab KO models, adding another argument for the difference between these KO strains. Finally, all the minor points were addressed.
Reviewer 2 Report
The paper entitled “Genetic and functional differences between duplicated 2 zebrafish genes for human SCN1A” is an interesting study and the authors have collected a unique dataset using cutting edge methodology. The paper is generally well written and structured. The research group generated scn1Laa knock out zebrafish and compared its phenotype to scn1Lab knockouts discovering that the genetic and functional differences can have implications for the use of zebrafish as a model for Dravet Syndrome. It is clear that the study is a continuation of the research group's.
I recommend the publication of this original article in cells
Author Response
Dear Reviewer 2,
We thank you for your reviewing our manuscript. The revised version is uploaded today.
Round 2
Reviewer 1 Report
The responses are satisfactory. The manuscript is of interest and well structured.